# Effect of Chronic Tibolone Administration on Memory and Choline Acetyltransferase and Tryptophan Hydroxylase Content in Aging Mice

**DOI:** 10.3390/brainsci14090903

**Published:** 2024-09-06

**Authors:** Tzayaka Castillo-Mendieta, Guadalupe Bautista-Poblet, Angélica Coyoy-Salgado, Emily L. Castillo-García, Rodolfo Pinto-Almazán, Claudia Erika Fuentes-Venado, Teresa Neri-Gómez, Christian Guerra-Araiza

**Affiliations:** 1CONAHCyT-Unidad de Investigación Médica en Enfermedades Neurológicas, Hospital de Especialidades, Centro Médico Nacional Siglo XXI, Instituto Mexicano del Seguro Social, Av. Cuauhtémoc 330, Mexico City C.P. 03940, Mexico; 2Unidad de Investigación Médica en Farmacología, Hospital de Especialidades, Centro Médico Nacional Siglo XXI, Instituto Mexicano del Seguro Social, Av. Cuauhtémoc 330, Mexico City C.P. 06720, Mexicoacoyoys@gmail.com (A.C.-S.);; 3Ciencias Biológicas y de la Salud, Universidad Autónoma Metropolitana, Mexico City C.P. 09340, Mexico; 4Sección de Estudios de Posgrado e Investigación, Escuela Superior de Medicina, Instituto Politécnico Nacional, Plan de San Luis y Díaz Mirón, Mexico City C.P. 11340, Mexico; rodolfopintoalmazan@gmail.com (R.P.-A.);; 5Servicio de Medicina Física y Rehabilitación, Hospital General de Zona No 197 IMSS, Texcoco C.P. 56108, Mexico; 6Laboratorio de Patología Molecular, Unidad de Investigación Biomolecular en Cardiología, Hospital de Cardiología, Centro Médico Nacional Siglo XXI, Instituto Mexicano del Seguro Social, Av. Cuauhtémoc 330, Mexico City C.P. 03940, Mexico

**Keywords:** aging mice, tibolone, memory, hippocampus, tryptophan hydroxylase, choline acetyltransferase

## Abstract

Gonadal steroids exert different effects on the central nervous system (CNS), such as preserving neuronal function and promoting neuronal survival. Estradiol, progesterone, and testosterone reduce neuronal loss in the CNS in animal models of neurodegeneration. However, hormone replacement therapy has been associated with higher rates of endometrial, prostate, and breast cancer. Tibolone (TIB), the metabolites of which show estrogenic and progestogenic effects, is an alternative to reduce this risk. However, the impact of TIB on memory and learning, as well as on choline acetyltransferase (ChAT) and tryptophan hydroxylase (TPH) levels in the hippocampus of aging males, is unknown. We administered TIB to aged C57BL/6J male mice at different doses (0.01 or 1.0 mg/kg per day for 12 weeks) and evaluated its effects on memory and learning and the content of ChAT and TPH. We assessed memory and learning with object recognition and elevated T-maze tasks. Additionally, we determined ChAT and TPH protein levels in the hippocampus by Western blotting. TIB administration increased the percentage of time spent on the novel object in the object recognition task. In addition, the latency of leaving the enclosed arm increased in both TIB groups, suggesting an improvement in fear-based learning. We also observed decreased ChAT content in the group treated with the 0.01 mg/kg TIB dose. In the case of TPH, no changes were observed with either TIB dose. These results show that long-term TIB administration improves memory without affecting locomotor activity and modulates cholinergic but not serotonergic systems in the hippocampus of aged male mice.

## 1. Introduction

Cognitive decline occurs in adulthood, but it is accentuated as steroid hormones (estrogens and androgens) decline, showing a risk factor for neurodegenerative diseases, such as Alzheimer’s disease (AD) [1,2,3,4]. Several studies suggest that estrogen treatment has beneficial effects on different types of memory, such as verbal, spatial, and figural memory, and also attenuates cognitive decline in menopausal and postmenopausal women [5,6,7,8,9,10].

No significant loss of neurons characterizes aging-stage brains; however, there are notable changes in the number, diameter, length, branching, and density of dendritic spines [11]. The aging process is characterized by decreased neurotransmitters like acetylcholine (ACh), dopamine, and serotonin (5HT) [12,13,14]. Although these changes are intensified in dementia cases, most neurological changes occur in older adults [15].

As the brain is sensitive to androgens, cognitive function declines as men age due to decreased testosterone levels [16,17]. The decline in gonadal steroids due to aging impacts brain function, particularly memory, and is associated with the natural reduction in estrogens. These deficits may be linked to how estradiol affects cholinergic innervations in the hippocampus, a crucial brain structure for memory processes [18].

Using silastic-capsule implants or daily hormone injections, continuous estradiol administration improves spatial working memory, spatial reference memory, and memory for non-spatial objects in aged female rats [19,20,21] and mice [22,23,24]. Similarly, continuous estrogen administration in aged female rodents exerts numerous effects on the brain structures involved in learning and memory, such as the hippocampus. In aged female mice, estradiol increases hippocampal levels of synaptophysin—a presynaptic protein and indicator of synaptic plasticity [25].

In a study of male rodents castrated at birth, researchers found that their learning curves in the aquatic maze were similar to those of females. Conversely, testosterone produced learning curves similar to males when administered to newborn female rodents [26]. Other reports have described that testosterone, but not 17β-estradiol, improves spatial memory [27,28]. In contrast, other studies reported improved spatial memory after estrogen replacement [29,30] but not after administration of dihydrotestosterone, which can be converted into 17β-estradiol. Assessing the effects of different androgens can help distinguish between androgen-dependent and estrogen-dependent pathways. However, due to a lack of large-scale studies and long-term trials evaluating the benefits and risks of hormone replacement therapy (HRT) in men, there is still controversy surrounding the indication for testosterone supplementation [31,32]. Recent studies suggest that men with AD show lower testosterone levels, but testosterone treatment improves their cognition [33,34]. However, chronic administration of estrogens (alone or combined with progestogens) as HRT has been linked to side effects, including endometrial and breast cancer, as well as cardiovascular events [35,36]. Tibolone (TIB) is widely prescribed to treat menopausal symptoms and prevent bone loss as it selectively regulates estrogenic tissue activity [37]. TIB is metabolized into three biologically active metabolites: 3-α-hydroxy-TIB and 3-β-hydroxy-TIB (both with estrogenic effects), and Δ4-TIB, which exhibits progestogenic and androgenic effects [38]. Tissue-specific desulfation is necessary to activate TIB metabolites [39,40].

In premenopausal women undergoing treatment for uterine leiomyoma, TIB administration improved cognitive impairment caused by leuprolide acetate and enhanced mood and quality of life [41]. In hypothalamic neurons, TIB administration rapidly attenuated the γ-aminobutyric acid B receptor response [42]. In ovariectomized (OVX) rats, TIB reduced lipoperoxidation and increased antioxidant capacity in the cortex and hippocampus [43] and improved memory in an inhibitory avoidance task [44]. Another study found that female rats treated with TIB showed stronger staining for GFAP and c-fos in the cerebral cortex and hippocampus compared to control animals [45]. Our research group showed that TIB modulates the expression of cytoskeletal proteins, such as Tau, affecting neuronal plasticity in the cerebellum and hippocampus [46]. Moreover, it has been found that TIB estrogenic effects play a significant role in enhancing cognitive performance in OVX rats [47].

A previous study in aged male mice revealed that TIB administration influences neuronal plasticity by regulating Tau, GSK3/Akt/PI3K pathway, and CDK5 p35/p25 complexes in the hippocampus. This regulation is TIB concentration-dependent, with CDK5 content increasing at 0.01 mg/kg but decreasing at 1 mg/kg TIB dose [48]. Notably, the effect of TIB administration on memory and learning and ChAT and TPH content in aging males has not been previously explored. Therefore, this study aimed to fill this gap by evaluating the impact of long-term TIB administration on memory and learning and ChAT and TPH hippocampal content in a male murine aging model.

## 2. Materials and Methods

### 2.1. Animals

Male C57BL/6J mice at two different age stages (3 months and 18 months of age) were housed per group and maintained with a 12 h light-dark cycle (lights on at 9:00 p.m.) and provided with water and food ad libitum. Experimental procedures were performed in agreement with the Science and Ethics Committee of the Mexican Social Security Institute (IMSS) with registration number R-2012-785-077 and adhered to the guidelines established by the Mexican Ministry of Agriculture and Livestock (NOM-062-Z00-1999, SAGARPA) and the Guide for the Care and Use of Laboratory Animals of the National Research Council. Every effort was made to reduce animal suffering and the number of animals in each experiment.

### 2.2. Treatments

Aged mice (18-month-old) were randomly divided into four groups (*n* = 9 per group). The first group was not treated, while the remaining three groups received daily one of the following treatments: (1) vehicle (Veh) (water), (2) 0.01 mg/kg TIB, or (3) 1.0 mg/kg TIB by oral gavage for 12 weeks. Untreated young intact mice (3-month-old) were used as controls.

### 2.3. Memory Evaluation

We used a white-painted plastic cubic arena (40 × 40 × 40 cm) for the object recognition memory task. We used two arenas with different textures for the object-in-context recognition memory task. One was a white-plastic cubic arena, while the other was a black-plastic cylindrical arena (60 cm diameter and 40 cm height). Both arenas had cardboard figures on their walls, serving as spatial landmarks. Sawdust was used to cover the floor of both arenas. The arenas were placed in a dimly lit illuminated room. Tests were recorded with a video camera positioned above the arenas. The objects to be distinguished were black cubes and white balls. These objects were firmly attached to the floor with Velcro to prevent displacement at the back corners of the arena, 10 cm from the walls. Furthermore, the objects were thoroughly cleaned with 70% ethanol, and the sawdust was stirred after each trial to avoid any olfactory cues.

Animals were transported from the vivarium to the experimental room one hour before each session and kept in this room for an additional hour after each session to avoid any stressful conditions affecting their performance or task consolidation. The contexts, objects, and their relative positions were counterbalanced for all experiments. All experiments were carried out in independent groups. We defined exploration as either pointing the nose toward an object less than 1 cm away or touching the object with the nose. Turning around or sitting on the objects was not considered an exploratory behavior. Mice with a total exploration time of <10 s in the training or testing phases were excluded from further analysis.

#### 2.3.1. Object Recognition Task

For three consecutive days, mice were manipulated for 1 min and placed in the cubic arena without objects for 3 min. During the sample phase, animals were placed in the arena facing the wall opposite the objects and allowed to explore two identical objects (either two cubes [A1] or two balls [A2]) for 5 min. Memory was tested either after 1 h (short-term memory) or 24 h (long-term memory). During the memory test, mice were presented with one of the previously seen objects (familiar object, A3) and a new object (B) and allowed to explore freely for 3 min. Short-term memory (STM) and long-term memory (LTM) were evaluated in all groups. Each animal’s test was videotaped for later analysis. Two experimenters, blind to treatment conditions, manually recorded the mice’s behavior and analyzed each video. We calculated the object recognition index as follows: time of exploration of novel object/(time of exploration of familiar object + time of exploration of novel object) [49]. A recognition index of 0.5 reflected no preference for any object, and an index higher than 0.5 indicated a preference for novel objects.

#### 2.3.2. Object-in-Context Recognition Task

For three consecutive days, mice were manipulated for 1 min. Immediately after that, the animals were habituated to each of the two object-free contexts for 3 min. A period of 90 min was allowed to elapse between the habituation of one context and the other. In the first sample phase, mice were placed in one of the arenas (context 1) facing the wall opposite the objects. They were given 10 min to explore two different objects, a cube (A1) and a ball (B1). Phase 2 of the experiment was conducted the following day. Mice were placed in the other arena (context 2) with two identical objects, two cubes (A2) or two balls (A3), replicas of those previously presented. The mice were given 10 min to explore the objects. Memory was evaluated after 60 min (STM) or 24 h (LTM). For the memory test, mice were reintroduced to context 2 and allowed to explore two objects. One object was a replica of an object presented in context 2 during phase 2 (A4), and the other object was a replica of an object previously presented in context 1 but not presented in context 2 (B2). Mice were allowed to explore these objects freely for 3 min.

Context recognition indexes were calculated by dividing the time of exploration of a familiar object in a novel context by the sum of the time of exploration of the familiar object in the novel context and the time of exploration of the familiar object in the familiar context. A recognition index of 0.5 reflected no preference for any object. An index value higher than 0.5 indicated a preference for familiar objects in a new environment. In this protocol, one of the objects is presented twice before the retention test. In contrast, the other object is only shown once, which may make one object more familiar.

#### 2.3.3. Elevated T-Maze

The elevated T-maze (ETM) is a test based on ethology that has been used to investigate the impact of anxiolytic drugs on memory and the relationship among the neural systems involved in such modulation. This test measures two types of aversive-motivated behaviors in mice: learned (or conditioned) anxiety, represented by inhibitory avoidance behavior, and innate (or unconditioned) fear, expressed by one-way escape [50,51,52].

We followed the methodology documented by De-Mello and Carobrez [53] using an ETM comprising three arms of equal dimensions (30 cm × 5 cm) elevated 30 cm above the floor. One of the arms was enclosed by lateral walls (20 cm high) and positioned perpendicularly to the opposite open arms. Mice’s behavior was recorded on video during each trial.

On the day of training, mice were placed at the end of the enclosed arm of the T-maze, facing the intersection of the arms, and allowed to explore the area. The trial concluded when the animal placed its four paws on one of the open arms or remained on the enclosed arm for 300 s (avoidance criterion). During the 30 s interval between trials, animals were returned to their cage, and the maze was cleaned with a 10% *v*/*v* ethanol solution. On the same day, each animal was again placed at the end of the enclosed arm with as many attempts as necessary until it remained for 300 s in the enclosed arm of the maze. An attempt or try was considered if the animal extended its body from the enclosed arm and placed one, two, or three paws on one of the open arms before returning to its original position. The risk index was calculated as follows:Risk index (RI) = frequency of tries/open-arm entry + frequency of tries,
in which an RI value of 1 represented no successful open-arm entry, and an RI of 0 indicated that every movement toward the open arm was successful.

### 2.4. Spontaneous Activity Test

We assessed spontaneous mice activity in an automated counter to prevent inaccurate task results due to motor function changes. The counter comprised an acrylic cage (51.1 × 9.5 × 69.2 cm) with two arrays of 15 infrared beams positioned perpendicularly. An array of photocells above the animal detected the vertical activity. The beams were spaced 2.5 cm apart, and the interruption of each beam generated an electrical impulse. This impulse was then processed and presented as a registered movement or count using the Opto-Varimex system (Columbus Instruments, Columbus, OH, USA). Total activity over a 5 min session was recorded after the T-maze task [47].

### 2.5. Protein Extraction and Western Blotting

Twenty-four hours after the last TIB administration (at 10 a.m.), mice were sacrificed by decapitation. The brain was removed, and the hippocampus was dissected according to the Atlas of Paxinos and Watson [54]. This region was immediately processed for protein extraction. The hippocampus was homogenized in lysis buffer containing protease inhibitors (10 mM Tris–HCl, 1 mM dithiothreitol, 30% glycerol, 1% Triton X-100, 15 mM sodium azide, 1 mM EDTA, 4 μg/mL leupeptin, 22 μg/mL aprotinin, 1 mM PMSF, and 1 mM sodium orthovanadate). Proteins were obtained by centrifugation at 12,700× *g* at 4 °C for 15 min and quantified according to the Bradford method (Bio-Rad, Hercules, CA, USA). Proteins (50 μg) were separated by electrophoresis on 10% SDS–PAGE gel at 80 V and transferred onto Immobilon–P transfer membranes (Millipore, Bedford, MA, USA) at 20 V at room temperature for 30 min. Membranes were blocked with Odyssey Blocking Buffer (LICOR) and incubated at 4 °C overnight with primary antibodies (diluted at 1:1000). Pre-stained broad-range markers (Biorad, Hercules, CA, USA) were included for size determination.

We used the following antibodies: goat anti-ChAT polyclonal antibody (Ab-144P; Merck, Burlington, MA, USA), mouse anti-TPH monoclonal antibody (T0678; Merck, Burlington, MA, USA), and GAPDH (MAB374; Merck, Burlington, MA, USA) at a dilution of 1:1000. After incubation with primary antibodies, membranes were washed and incubated with horseradish peroxidase-coupled secondary antibodies (Santa Cruz Biotechnologies, Santa Cruz, CA, USA) diluted at 1:15,000. Highly precise detection of proteins from Western blots was performed using an enhanced chemiluminescence system (Millipore, Billerica, MA, USA). The intensity of each band was quantified by densitometry (Hewlett Packard, Palo Alto, CA, USA), and densitometric analysis was conducted with KODAK 1D Image Analysis Software version 3.6 (Eastman Kodak, Rochester, NY, USA). The density of each band, as well as the density of the mean of all groups, were normalized to its respective loading control (GADPH). To minimize inter-assay variations, we processed the samples from all animal groups in each experiment concurrently and under the same conditions.

### 2.6. Statistical Analysis

Data were analyzed using Student’s *t*-test and ANOVA followed by the Tukey post hoc test. Probability values were calculated using the GraphPad Prism version 2.01 software program (GraphPad, CA, USA). *p*-values of <0.05 were considered statistically significant.

## 3. Results

### 3.1. Effect of Aging on Cognitive Ability

In the object recognition task, the recognition index (RI) was calculated to evaluate STM and LTM exhibited by young and aged animals (Figure 1A) (Appendix A). Data analysis showed a significantly lower RI value in the aged group than in the young group (1 h, t = 4.111; *p* < 0.01; 24 h, t = 8.006; *p* < 0.01) for both types of memory. In the object-in-context recognition task (Figure 1B), the RI was also calculated to evaluate memory in young and aged mice. When analyzing the data, a significant difference in RI values was found for STM and LTM. These RI values were lower in the aged group at both times (1 h, t = 6.568; *p* < 0.01; 24 h, t = 11.50; *p* < 0.05).

We further analyzed memory with the ETM. Data analyzed exhibited statistically significant differences between the groups in latency in entering one of the open arms (Figure 2). The aged group showed a lower latency value than the young animals (t = 3.278; *p* < 0.01) (Figure 2A). Furthermore, we observed a statistically significant decrease in the RI-value in the aged group compared to the young group (t = 2.427; *p* < 0.05) (Figure 2B).

The results of spontaneous activity are presented as the mean ± standard error (SE) of the number of counts in 5 min (Figure 3). Data analysis indicated that except for total activity (t = 1.777; *p* = 0.085), vertical activity (t = 2.209; *p* < 0.05), and ambulatory activity (t = 3.745; *p* < 0.01) decreased significantly in aged mice.

### 3.2. Effect of Long-Term TIB Treatment on Cognitive Ability in Aging Mice

Once we analyzed the effects of aging, we evaluated the effect of long-term TIB treatment on cognitive ability. Figure 4A shows the recognition index at 1 and 24 h in the object recognition task. Upon performing statistical analysis, a significant difference was observed between the control group (Veh) and TIB-treated groups at both evaluation times (1 h, F2,24 = 18.09; *p* < 0.01; 24 h, F2,24 = 27.57; *p* < 0.01). It was also noted that both doses of TIB increased the preference for the novel object in the test phases 1 and 24 h after the recognition session.

The effect of long-term TIB treatment on the novel object-in-context exploration 1 and 24 h after exposure to the familiar objects is shown in Figure 4B. The statistical analysis revealed a difference between TIB-treated groups in STM and LTM (1 h, F2,24 = 31.94; *p* < 0.01; 24 h, F2,24 = 25.58; *p* < 0.01). We also noted that TIB-treated groups showed a higher preference for the novel object than the vehicle group.

Long-term TIB treatment using the ETM (Figure 5A) demonstrated a significant increase in latency on entering one of the open arms (F2,24 = 57.35; *p* < 0.01) with both TIB doses. In contrast, no significant difference was observed for the risk index (F2,24 = 0.994; *p* > 0.05) between controls and TIB-treated groups (Figure 5B).

Figure 6 exhibits the effect of long-term TIB treatment on spontaneous activity in aged mice. Statistical analysis indicated that TIB treatments did not modify the locomotor activity of the mice (ambulation, F2,24 = 2.469; *p* > 0.05; vertical activity, F2,24 = 0.4796; *p* > 0.1). However, the post hoc analysis indicated a significant decrease in total activity with the TIB 0.01 mg/kg dose compared with the vehicle (F2,24 = 4.122; *p* < 0.05).

### 3.3. ChAT and TPH Content in the Hippocampus of Aging Mice

We observed a statistically significant difference in the hippocampal levels of ChAT between aged and young mice (Figure 7). In contrast, no statistically significant differences were found between the TIB-treated mice and the control group. However, we observed a declining trend in ChAT content compared to untreated aged mice. Upon analyzing the results obtained for TPH levels in the hippocampus, no statistically significant differences were found between the old and young mice groups nor between the groups administered with different doses of TIB (0.01 and 1.0 mg/kg) and the group administered with Veh (Figure 8).

## 4. Discussion

As men age, testosterone levels decrease, resulting in symptoms such as asthenia, decreased muscle mass, osteoporosis, and reduced sexual activity. This set of symptoms is known as androgen deficiency in the aging male (ADAM) syndrome [55,56]. However, several studies question the use of HRT due to its association with an increased risk of prostate cancer and stroke. Although this risk could be decreased by using TIB (a synthetic hormone with estrogenic, progestogenic, and androgenic effects), its effect on cognitive impairment associated with aging is unknown.

In the present study, we evaluated how chronic administration of TIB affects old mice’s cognitive capacity (memory and learning). We analyzed how different doses of TIB affect memory consolidation in the object and object-in-context recognition tasks. These tasks helped distinguish important components of recognition memory, such as the object’s identity and the context where it was found [57].

Our results showed differences between young and old mice. Memory retention and consolidation are mostly affected in aged mice, probably because of the neurodegeneration of different brain regions in these animals [58]. Although recognition of prior information remains relatively stable throughout life, the ability to encode and recall contextual information tends to decline with age [59]. When TIB was administered to the aged mice group, we observed increased object and object-in-context recognition indexes, suggesting improved memory consolidation. These results indicate that TIB exerts an effect on neurodegenerative processes. TIB is a neuroendocrine modulator [60] and a neurotransmitter modulator [47]. Moreover, studies have shown that TIB decreases the formation of paired helical filaments (PHFs) by increasing the amount of dephosphorylated Tau [46].

Regarding memory and learning, aged mice showed a significant decrease in latency compared to young mice in the ETM, suggesting their inability to learn the passive inhibitory avoidance response. This difference may be due to several reasons, such as decreased locomotor activity, lack of motivation to explore, muscle mass loss, worse energy homeostasis, and poorer sensory abilities, among other factors related to old age. However, the risk index value in both groups was close to 1, suggesting an ability to acquire information about their environment and the passive inhibitory avoidance response. Conversely, we observed a significant increase in latency in mice treated with TIB (both doses), indicating that aged mice learned the passive inhibitory avoidance response. In these groups, no changes were observed in the risk index value, suggesting that, while the animals were in the closed arm, there was no lack of motivation to explore nor any alteration of locomotor activity. These results confirm previous findings that show reduced CNS plasticity in aged animals associated with learning [60,61,62]. Emotional processes may also be affected at this stage of life, possibly due to substantial changes that the brain undergoes throughout life, from which the hippocampus [63,64,65] and the amygdala [66] are not exempt.

Furthermore, the loss of dopaminergic neurons in the substantia nigra due to aging could originate a reduction in vertical and ambulatory activity in aged mice [67]. Additionally, we observed a decrease in total activity with the 0.01 mg/kg TIB treatment. Despite this finding, data analysis indicated that the animals’ performance in memory and learning tests was unaffected. Several studies have reported that castrated male rats show decreased spatial working memory [68,69,70,71,72] and poor performance in the radial arm maze, T-maze, and object recognition tasks due to drastically decreased testosterone levels [73]. However, testosterone administration restored spatial working memory [33,74,75].

Aging reduces the baseline concentrations of some neurotransmitters, such as ACh, NA, and 5-HT [76,77]. Cholinergic neurons in the diagonal band-cingulate cortex, medial septal nucleus (MS), and the nucleus basalis magnocellularis (NBM) influence learning, memory, and attention [78,79,80]. Cholinergic innervation in the hippocampus and cerebral cortex is primarily provided by these neurons, which are severely impacted by aging and AD [81,82]. Drugs that reduce cholinergic activity cause cognitive problems similar to aging and AD, while drugs that improve cholinergic activity help with cognitive function in healthy individuals and reduce AD-related cognitive decline [81,83]. Moreover, old animals showed poor water-maze performance, decreased baseline ACh release concentrations, and reduced serotonin-agonist-induced ACh release [77].

In the present work, we observed increased ChAT concentrations in old mice compared to young mice. Therefore, we need to clarify the results of previous studies performed in aged C57BL/6J mice regarding ChAT content. Some studies indicate that ChAT content decreases, but others show that it is not altered, and others report that it increases [25]. Moreover, increased estrogen receptor expression, particularly estrogen receptor beta (ERβ), has been observed in aged animals [84]. Similarly, estradiol has been reported to increase ChAT expression in rodent hippocampus [85]. These results could explain our findings regarding increased ChAT expression in old mice relative to young mice. Thus, increased ERβ expression could influence the increased ChAT content in aged animals, although further research is required to confirm these observations.

The deterioration of other cholinergic characteristics associated with aging, including muscarinic binding, ACh uptake, and a decline in mnemonic memory, may be compensated for by this action [22]. When TIB (0.01 and 1.0 mg/kg) was administered to old mice, no statistically significant differences were observed; however, a downward trend was noted in the groups administered with TIB compared to Veh. Given that E2 has been shown to increase acetylcholine release [86] and that an M2 muscarinic receptor antagonist blocks at least some of the effects of E2 on hippocampal excitatory synaptic function, this effect may be caused by the estrogenic metabolite of TIB [29]. Because female estrogen receptor beta deletion mice spend less time examining the novel item, the impact of TIB might be caused by the activity of its estrogenic metabolite [87]; thus, the role of this receptor in recognition memory is important and a possible explanation for the mechanism of action of TIB. According to some studies, TIB administration decreased lipoperoxidation (LPO), avoided neuronal death, improved cholinergic deficiency, and lessened cognitive and motor impairment in the oxidative stress (OS) model induced by O3 exposure [88,89].

Research on post-mortem human and animal brains found no evidence of a decline in 5-HT or its main metabolite, 5-hydroxyindolacetic acid (5-HIAA) concentrations with aging. TPH activity is not altered by age; however, there is occasionally an age-related rise in the brain concentration of these substances [79,90]. Aged rats exhibit abnormal serotonergic fibers, mainly in the hippocampus CA1 and dentate gyrus molecular layer. It is hypothesized that the abnormal 5-HT fiber shape may represent local hippocampus serotonergic afferent degradation with aging [91]. Upon analyzing the results of TPH concentrations, no differences were found between old and young mice; this same trend persisted when TIB was administered to old mice, for which no statistically significant differences were observed between old mice administered with TIB and their respective controls. Evidence shows that estrogen regulates serotonergic neurotransmission via altering the activity of monoamine oxidase, TPH, serotonin reuptake transporters, or the number of serotonergic receptors [92]. High doses of estradiol and TIB increase THP concentrations, suggesting that TIB exerts an estrogenic effect on TPH regulation [47,92]. Although the activation of serotonergic receptors in the present study did not increase TPH concentrations, other authors have hypothesized that depending on the receptor subtype, the affected brain regions, and the drug dose, serotonergic receptor activation may result in anxiogenic or anxiolytic responses [93].

According to many reports, TIB treatment impacts CNS neurodegenerative processes. It has also been suggested that TIB exerts neuroendocrine and Tau neurotransmitter-modulating properties [44,46,94,95,96]. The effective use of TIB as a protective agent against lipid/protein oxidative effects induced by ozone exposure in rats has also been reported [84]. Our results suggest that, although cognitive changes are more severe in dementia, they can also be observed in healthy elderly individuals. Memory deficits in aged animals mainly affect the acquisition and consolidation of new information and the recall of this information (long-term memory). Although the mechanisms by which TIB exerts these effects are not yet well understood [47], it plays a crucial role in the CNS. Based on our findings, we suggest that long-term TIB treatment improves cognitive abilities in aged mice without affecting locomotor activity, possibly through the action of its androgenic and estrogenic metabolites.

## 5. Conclusions

Although the use of TIB as HRT and its effects on decreasing neurodegenerative processes in elderly men have not been investigated, in this work we observed that long-term TIB treatment improved memory in a murine model of aging. Therefore, further studies are required to elucidate the mechanisms of TIB and to determine the optimal HRT conditions for various groups.

## Figures and Tables

**Figure 1 brainsci-14-00903-f001:**
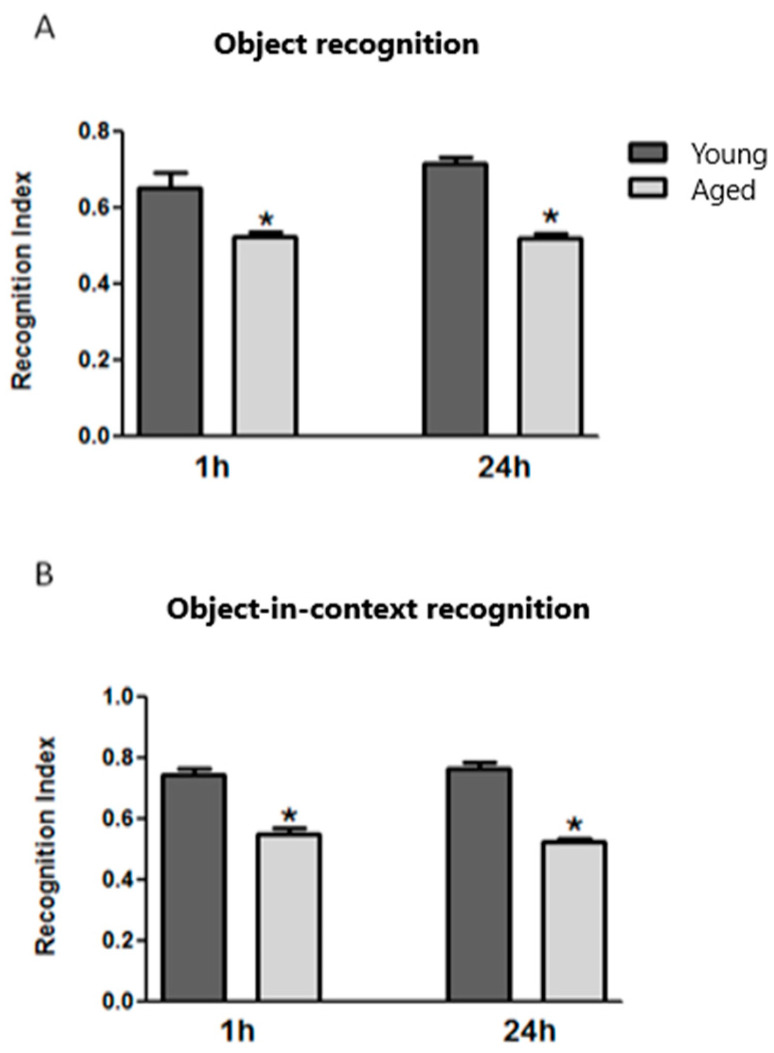
Effect of age on short-term (1 h) and long-term memory (24 h) in the object recognition (**A**) and the object-in-context recognition (**B**) tasks. Data represent the mean ± S.E. (n = 9). * *p* < 0.05 vs. young mice.

**Figure 2 brainsci-14-00903-f002:**
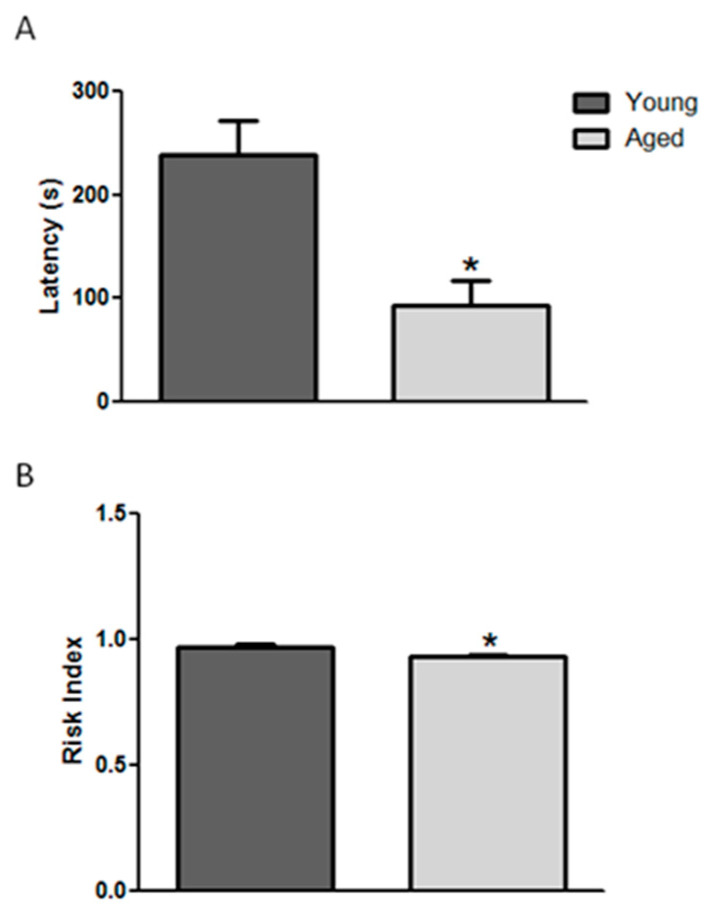
Effect of age on memory evaluated with the T maze task. Latency (**A**) and risk index (**B**) were calculated for each group (n = 9). Data represent the mean ± S.E. * *p* < 0.05 vs. young mice.

**Figure 3 brainsci-14-00903-f003:**
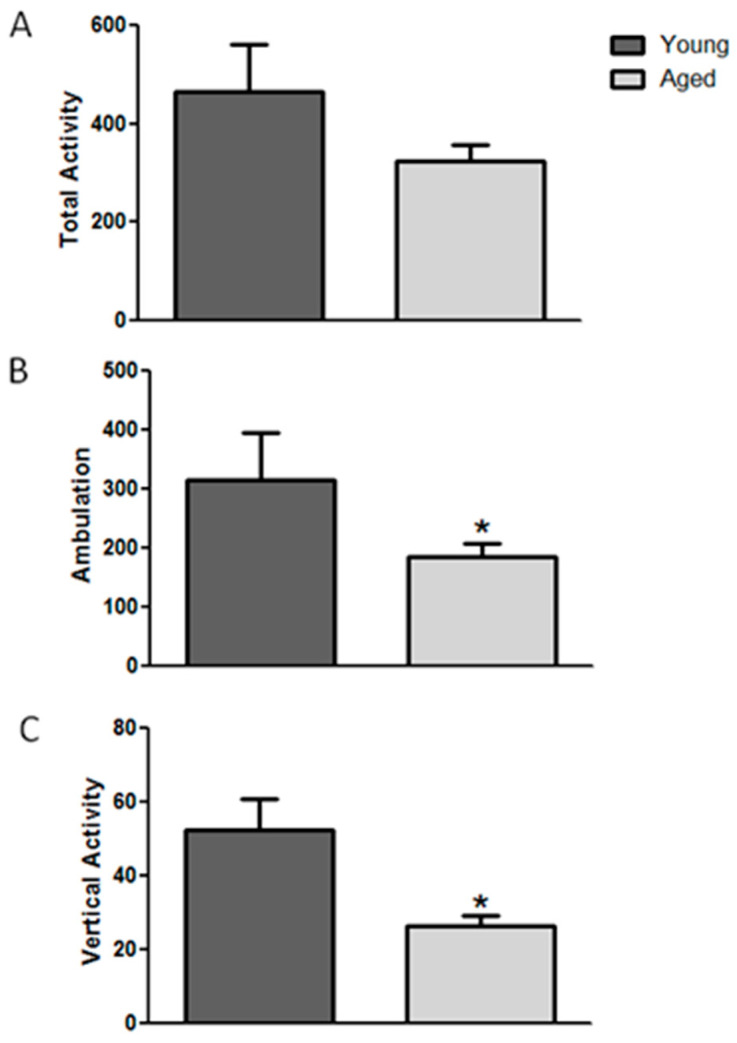
Effect of age on total activity (**A**), ambulation (**B**), and vertical activity (**C**) in young and aged mice (n = 9). Data represent the mean ± S.E. * *p* < 0.05 vs. young mice.

**Figure 4 brainsci-14-00903-f004:**
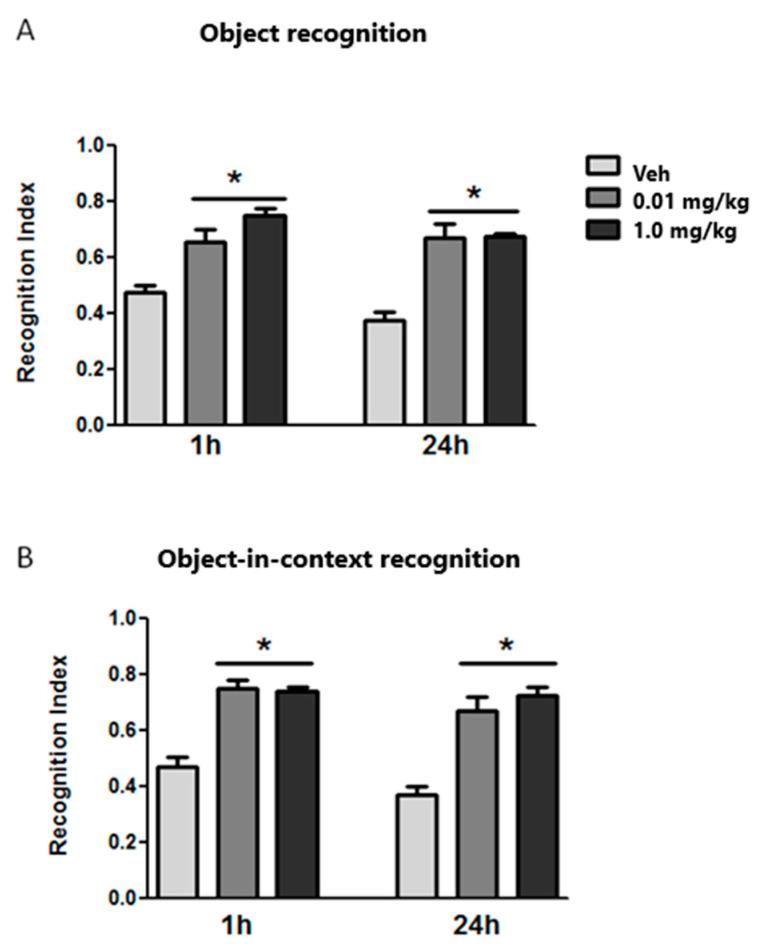
Effect of TIB on short-term (1 h) and long-term memory (24 h) in aged mice. (**A**) Object recognition and (**B**) object-in-context recognition tasks. Data represent the mean ± S.E. (n = 9). * *p* < 0.05 vs. Veh. Veh = vehicle.

**Figure 5 brainsci-14-00903-f005:**
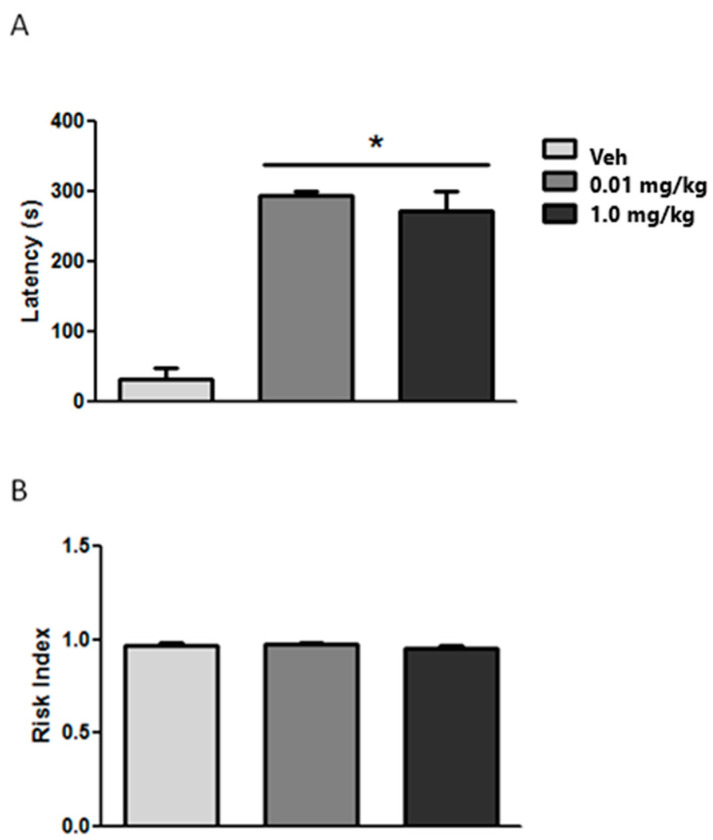
Effect of long-term TIB treatment on latency (**A**) and risk index (**B**) in the ETM task in aged mice. Data represent the mean ± S.E. (n = 9). * *p* < 0.05 vs. vehicle. Veh = vehicle.

**Figure 6 brainsci-14-00903-f006:**
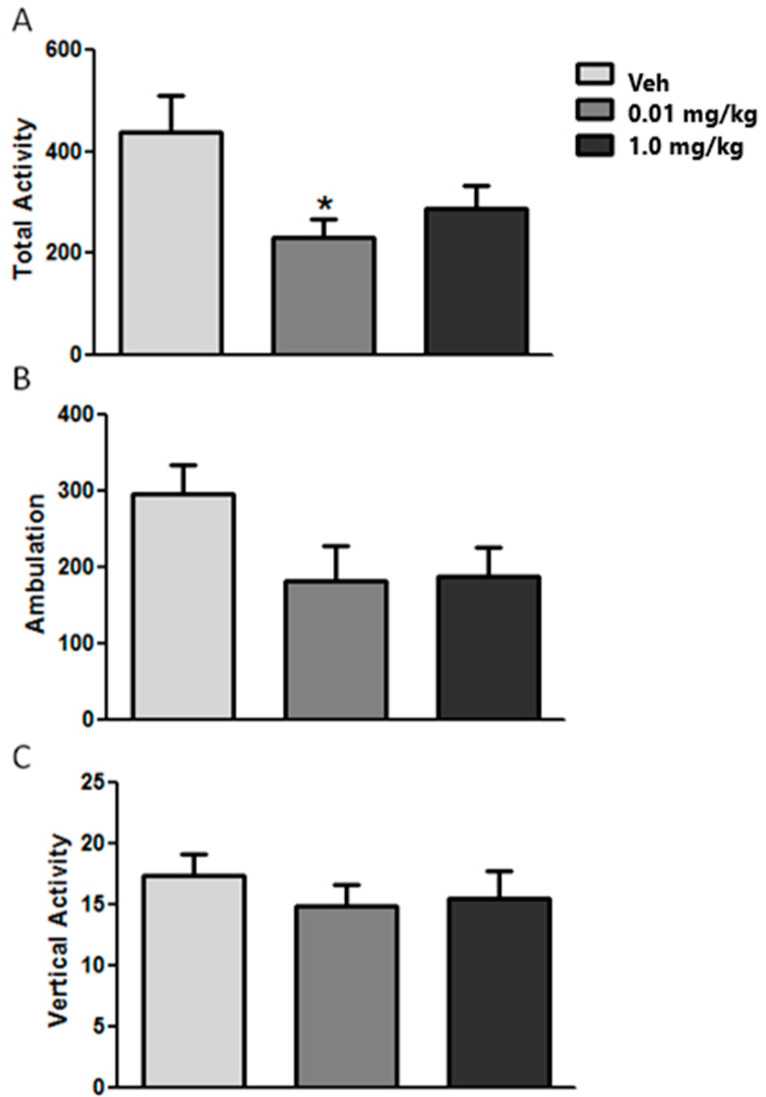
Effect of long-term TIB treatment at different doses on spontaneous activity in aged mice: (**A**) total activity, (**B**) ambulation, and (**C**) vertical activity. Data represent the mean ± S.E. (n = 9). * *p* < 0.05 vs. vehicle. Veh = vehicle.

**Figure 7 brainsci-14-00903-f007:**
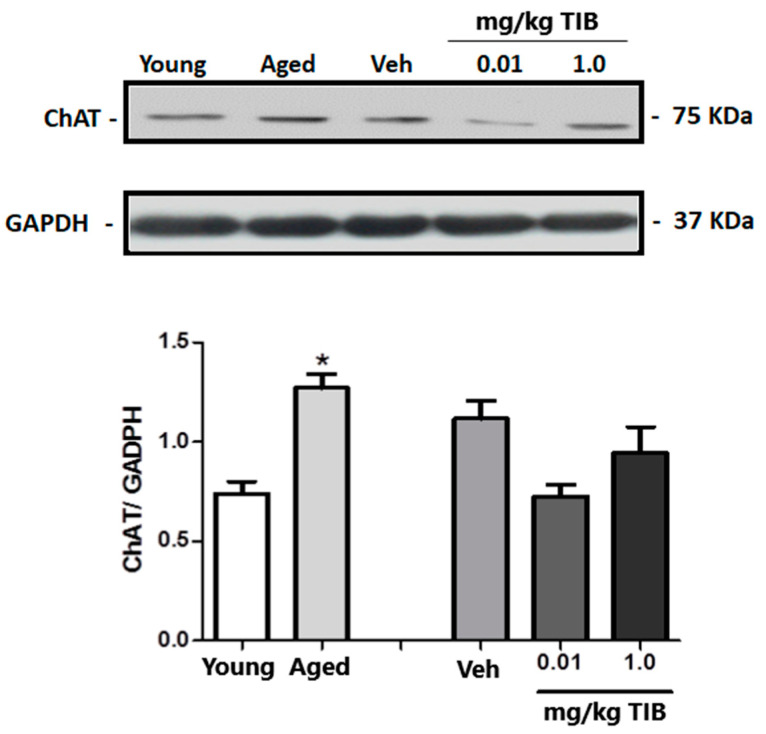
ChAT content in the hippocampus of intact young and aged mice and mice treated with TIB at different doses (0.01 and 0.1 mg/kg). Protein levels were quantified by Western blotting. Densitometric analysis was conducted and corrected using GADPH content data. Results are expressed as mean ± S.E. (n = 6). * *p* < 0.05 vs. young.

**Figure 8 brainsci-14-00903-f008:**
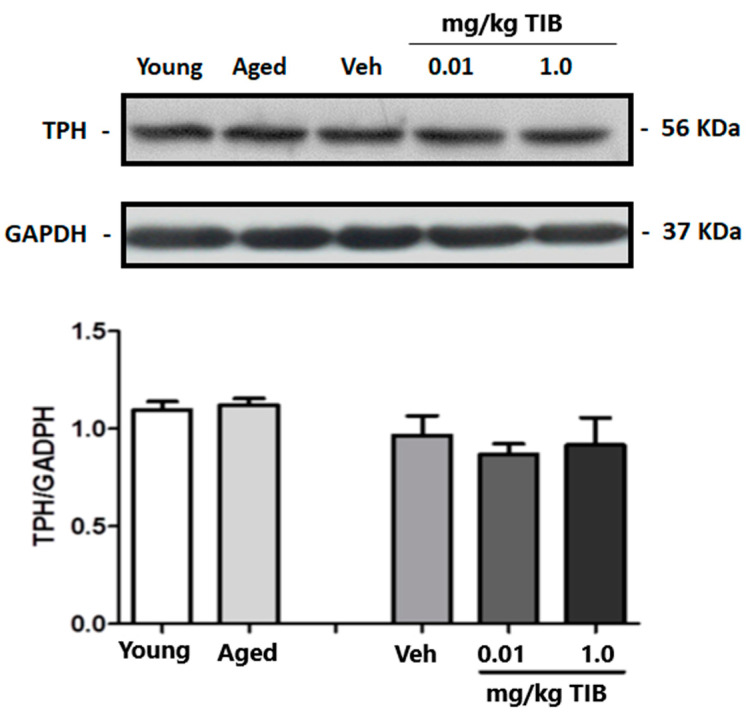
TPH content in the hippocampus of intact young and aged mice and mice treated with TIB at different doses (0.01 and 0.1 mg/kg). Protein levels were quantified by Western blotting. Densitometric analysis was conducted and corrected using GADPH content data. Results are expressed as mean ± S.E. (n = 6).

## Data Availability

All data generated or analyzed during this study are included in this article.

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
