# Peer review of "Effect of Chronic Tibolone Administration on Memory and Choline Acetyltransferase and Tryptophan Hydroxylase Content in Aging Mice"

_brainsci, 2024, doi:10.3390/brainsci14090903_

Round 1

Reviewer 1 Report

Comments and Suggestions for Authors

Steroids play a crucial role in maintaining and regulating the body’s function, mediating many important physiological processes, such as reproduction, sexual differentiation, ionic and carbohydrate homeostasis and stress responses. In a number of brain structures, these hormones are responsible for diverse neurochemical processes both during development and in adulthood. 

It has been a pleasure to evaluate this very timely and elegant research article by Castillo-Mendieta and colleagues. Authors show the first time that long-term tibolone administration improved memory without affecting locomotor activity and modulated cholinergic, but not serotonergic, signaling in the hippocampus of aged mice.

The behavioural and biochemical methods (WB) are in general very well considered and all procedures were kept the high standard.  Appropriate statistical methods were applied with sufficient number of experimental data. The study is perfectly documented and manuscript is distinctly informative. All figures and tables are also well designed and clear. To sum up, this article may be considered as a valuable contribution to the field of experimental neuroscience and applied psychopharmacology.

I have got  some suggestions for the Authors:

Methods

1. What is the rationale of tibolone doses (0.01 and 1.0 mg/kg/day) used in the study? Are there any references supporting this?

2. What kind of anaesthesia has been applied before the tissue preparation?

3. I could not find a basic characteristics of all primary antibodies used in the study (host, dilution, clonality, cross reactivity etc.)

4. Western blotting is a reliable method for this kind of study, however the experiment would significantly benefit from immunohistochemical analysis of hippocampal slices. Have Authors got some additional brain samples for IHC? If not, this should be taken into consideration in their future projects.

Comments on the Quality of English Language

OK.

Author Response

Reviewer’s comments, author responses, and manuscript changes

We thank the referees for carefully reviewing the manuscript and their opinions regarding its scientific content and presentation. 

Reviewer #1

  1. What is the rationale of tibolone doses (0.01 and 1.0 mg/kg/day) used in the study? Are there any references supporting this?

Response: Thank you for your observation. For clarification purposes, we have added the following paragraph in the introduction section (page 2, lines 102-106):

A previous study in aged male mice revealed that TIB administration influences neuronal plasticity by regulating Tau, GSK3/Akt/PI3K pathway, and CDK5 p35/p25 complexes in the hippocampus. This regulation is TIB concentration-dependent, with CDK5 content increasing at 0.01 mg/kg but decreasing at 1 mg/kg TIB dose [Neri-Gómez et al. 2017].

This paragraph refers to the regulation that tibolone exerts on some essential kinases in modulating neuronal plasticity. Based on this, we decided to conduct our experiments using these doses of tibolone.

  1. Neri-Gómez, J. Espinosa-Raya J, S. Díaz-Cintra, J. Segura-Uribe, S. Orozco-Suárez S, J. M. Gallardo, C. Guerra-Araiza, “Tibolone modulates neuronal plasticity through regulating Tau, GSK3β/Akt/PI3K pathway and CDK5 p35/p25 complexes in the hippocampus of aged male mice”, Neural Regen Res, 2017, vol. 12, no. 4, p. 588, doi: 10.4103/1673-5374.205098.
  2. What kind of anesthesia has been applied before the tissue preparation?

Response: As we analyzed the protein content in a brain region (hippocampus) in this study, we could not use any anesthetic drug that could interfere with the expression of these proteins. For this reason, the animals were sacrificed by decapitation. This information was specified in the Materials and Methods section (page 5, line 223).

  1. I could not find a basic characteristics of all primary antibodies used in the study (host, dilution, clonality, cross reactivity etc.)

Response: We apologize for missing this information. We have specified the information regarding the antibodies in the "2.5. Protein extraction and Western blotting" section (page 6, lines 237-239).

  1. Western blotting is a reliable method for this kind of study; however, the experiment would significantly benefit from immunohistochemical analysis of hippocampal slices. Have the Authors got any additional brain samples for IHC? If not, this should be taken into consideration in their future projects.

 Response: We greatly appreciate your suggestion. Immunohistochemical analysis of hippocampal slices would benefit our research, and we will consider this approach for future projects.

Reviewer 2 Report

Comments and Suggestions for Authors

I like the basic idea of the present study. The questionaire is clear, and the used procedures are adequate. I am a bit shocked by the behavioral data. The effects are very extrem (I never saw such effects and such low variability in object recognition before) and it would be important to known more about the way the behavior was quantified. Overall, the manuscript is not of sufficient quality and should be improved.

More specific comments:

(1) The manuscript appears to have been hastily prepared. It contains many small errors that make it difficult to read and understand. In general, the manuscript would also benefit from scientific language editing. Very often the sentences could be formulated more precisely and thus more scientifically. For specific examples, please see below.

(2) Abstract, line 34: There are a lot of tasks in the T-maze. Please specify. Then, the meaning of a latency (line 36) can also be evaluated.

(3) Abstract, line 36: "T-maze taste"

(4) Abstract, lines 40f: Transmission was actually not measured.

(5) Intro, lines 45-47: This sentence is hard to understand.

(6) Line 49: "prevent" - really?

(7) Line 56: What is the meaning of primarily" here?

(8) Line 58: "this hormone" - which hormone? The sentence before refers to androgens (a group of hormones).

(9) Line 72: What is "T"?

(10) Line 80: What is "To"?

(11) Line 92/100: What is "OVX"?

(12) Line 106: What are 129/C57BL/6 mice? Are this F1 hybrids of 129 (which subline) and C57BL/6?

(13) Line 107: I am not sure whether 18-months old mice can be named "aged male mouse model". Typically, aged mice are > 24 months.

(14) Lines 125/126: "...we used two arenas that had different physical textures. Two different arenas were used for the experiment." Also, the information on the sawdust covered floor is given twice.

(15) Line 133: "Black squares" or cubes?

(16) Lines 143ff: This led assume that the animals were observed (via monitor?) by an experimenter that manually score the behavior of the mice. Was this the case? Or was a video tracking system used for analyses, with a manual double-check? If manual analysis, was the experimenter blind to treatment conditions?

(17) Lines 162: "...two different contexts, which were 90 minutes apart,..."?

(18) Lines 190: "Carobrez52"

(19) Lines 196ff: This is very irritating. "The trial concluded when the mouse placed four paws on one of the open arms or stayed in the enclosed arm for a maximum of 300 seconds (avoidance criterion) after leaving the enclosed arm". However, two sentence later "When the mouse extended its body from the enclosed arm and placed one, two, or three paws on one of the open arms before returning to its original position, it was considered as a trial." So, these two definitions of a trial are not in agreement. Later, there is a formula with "tries". However, "tries" are not defined.

(20) Line 219: "Watson53"

(21) Figure 1: These data are unbelievable. With a group size of 9, there is more or less almost no variability (SE is SEM, correct?), also after 24h. Actually, it would be nice to see a table with the individual data (or a graph with paired exploration times for the two objects). I guess I never saw such robust data in object recognition.

(22) Line 256: " This value was lower in the aged group." Which value? The value in the sentence before was the differences between the two groups at the two time points.

(23) Line 261: ANOVA or t-test?

(24) Figure 5: Vehicle-treated mice are much worse than the untreated mice. There is actually a strong effect of the vehicle administrations, which is rescued by TIB. Can the authors give more details on the oral administrations? When exactly were they done? How much later were the test?

(25) Line 303: "Outpatient"?  (= Ambulation???)

(26) Figure 7/8: The blots strongly indicate that not only the protein expression levels of each animal was normalized to GADPH but also to the mean of all groups. Is this correct? If yes, this should be described. If my suggestion is not correct, why are the ratios to high? ChAT expression in aged mice is for example much weaker than GADPH expression. Nevertheless, the ratio is higher ca. 1.25 (which would indicate more ChAT expression than DADPH expression!).

(27) Lines 335ff: Somehow, the second sentence does not fit to the first.

(28) Lines 347ff: I got lost in this sentence: "The results obtained in the present work clearly show a difference between two age stages, where it was observed that the consolidation of memory is mostly affected in old mice, because in the object recognition and object-in- context tests, a decrease was observed in the group of old mice, this indicating a clear difference between the neurological states that two different age groups undergo, in which the neurodegeneration of different regions of the brain conditions a decrease in cognition in old mice in the recognition of objects- without-context". First, I guess that several parts of the sentence can be deleted since unnecessary. Second, dividing in 2 or 3 sentences would facilitate understanding. Third, why do you think it a problem of consolidation? It also could be retention, right?

(29) Line 263: Do mice really learn this avoidance response? Where is this shown? The data more look like this avoidance response is innate, as shown by risk indexes very close to 1.

(39) Line 378f: There are many potential reasons for decreased locomotor activity in aged mice. Loss of muscle mass, worse energy homeostatis, poorer sensory abilities, etc. could be also a potential reason.

(40) Line 379f: This sentence is incomplete. "In addition, long-term treatment with doses of 0.01 mg/kg"

(41) Line 384f: A verb is missing in this sentence. "Several studies have shown that castrated male rats impair spatial working memory [67–71], decreasing performance in the radial arm maze, in the T-maze, and in object recognition [72]."

(42) Line 386: I don't understand how castration can affect the context.

(43) Line 398: "In the present work, we observed that the ChAT concentration increased in old mice with respect to young mice. " Maybe discuss a possible explanation for this increase.

Comments on the Quality of English Language

See my comments and examples in the "specific comments".

Author Response

Reviewer’s comments, author responses, and manuscript changes

We thank the referees for carefully reviewing the manuscript and their opinions regarding its scientific content and presentation. 

Reviewer #2

I like the basic idea of the present study. The questionnaire is clear, and the used procedures are adequate. I am a bit shocked by the behavioral data. The effects are very extreme (I never saw such effects and such low variability in object recognition before), and it would be important to know more about the way the behavior was quantified. Overall, the manuscript is not of sufficient quality and should be improved.

More specific comments:

(1) The manuscript appears to have been hastily prepared. It contains many small errors that make it difficult to read and understand. In general, the manuscript would also benefit from scientific language editing. Very often the sentences could be formulated more precisely and thus more scientifically. For specific examples, please see below.

Response: Thank you for your observation. We have revised the scientific language and edited the entire manuscript.

(2) Abstract, line 34: There are a lot of tasks in the T-maze. Please specify. Then, the meaning of a latency (line 36) can also be evaluated.

(3) Abstract, line 36: "T-maze taste"

(4) Abstract, lines 40f: Transmission was actually not measured.

Response to observations 2-4: We apologize for sending an unpolished version of the abstract with several errors that the reviewer highlighted. We have corrected the errors and restructured the abstract (page 1, lines 27-42).

(5) Intro, lines 45-47: This sentence is hard to understand.

Response: Thank you for your observation. We clarified this sentence (page 1, lines 47-49).

(6) Line 49: "prevent" - really?

Response: We appreciate your observation. We have used a more appropriate word in this sentence to clarify the idea we wanted to communicate (page 1, line 51).

(7) Line 56: What is the meaning of primarily" here?

Response: Thank you for your observation. We clarified this sentence (page 2, lines 56-57).

(8) Line 58: "this hormone" - which hormone? The sentence before refers to androgens (a group of hormones).

Response: Thank you for your observation. We were referring to testosterone. We clarified this sentence (page 2, lines 58-59).

(9) Line 72: What is "T"?

Response: Thank you for your observation. We meant testosterone and changed T for “testosterone” throughout the manuscript.

(10) Line 80: What is "To"?

Response: We apologize for the typo. We also meant testosterone and changed To for “testosterone” (page 2, line 81).

(11) Line 92/100: What is "OVX"?

Response: We're sorry for missing this explanation. We meant ovariectomy (OVX), which we have already clarified in the text (page 2, line 93).

(12) Line 106: What are 129/C57BL/6 mice? Are this F1 hybrids of 129 (which subline) and C57BL/6?

Response: We apologize for this mistake in the mouse strain. We were referring to the C57BL/6J strain. We have already corrected this information throughout the manuscript.

(13) Line 107: I am not sure whether 18-months old mice can be named "aged male mouse model". Typically, aged mice are > 24 months.

Response: Thank you for your interesting observation. We evaluated two different technical reports by Jackson Laboratory. They report that for this mouse strain, old age is considered from 18 to 24 months, and animals older than 24 months are considered very aged. In addition, several studies have reported the analysis of genetic changes and biomarkers in 18-month-old animals.

Furthermore, we initiated TIB treatment on 18-month-old animals and continued the treatments for three months. Therefore, experiments were performed on 21-month-old animals, an age within the range considered as aged mice.

https://www.jax.org/news-and-insights/jax-blog/2017/november/when-are-mice-considered-old#

https://www.jax.org/research-and-faculty/research-labs/the-harrison-lab/gerontology/life-span-as-a-biomarker

Richard A. Miller. Biomarkers and the Genetics of Aging in Mice. In: Cells and Surveys, Should Biological Measures Be Included in Social Science Research? National Research Council (US) Committee on Population; Editors: Caleb E Finch, James W Vaupel, and Kevin Kinsella. Washington (DC): National Academies Press (US); 2001.

(14) Lines 125/126: "...we used two arenas that had different physical textures. Two different arenas were used for the experiment." Also, the information on the sawdust covered floor is given twice.

Response: We apologize for the redundancy. We have deleted the repeated information (page 3, lines 129-134).

(15) Line 133: "Black squares" or cubes?

Response: Thank you for your observation. We corrected the error by changing black “squares” to “cubes” (page 3, line 136).

(16) Lines 143ff: This led assume that the animals were observed (via monitor?) by an experimenter that manually score the behavior of the mice. Was this the case? Or was a video tracking system used for analyses, with a manual double-check? If manual analysis, was the experimenter blind to treatment conditions?

Response: Thank you for your comments. Each animal's test was videotaped for later analysis. Two observers analyzed each video, and each observer was blinded to the treatment conditions.

We have added this information in the methods section (page 4, lines 157-159).

(17) Lines 162: "...two different contexts, which were 90 minutes apart,..."?

Response: Thank you for your observation. We have rephrased this paragraph for a better understanding as follows:

For three consecutive days, mice were manipulated for 1 minute. Immediately after that, the animals were habituated to each of the two object-free contexts for 3 min. A period of 90 minutes was allowed to elapse between the habituation of one context and the other (page 4, lines 165-167).

(18) Lines 190: "Carobrez52"

Response: We apologize for the typo. We changed "Carobrez52" to Carobrez [53] (page 5, line 193).

(19) Lines 196ff: This is very irritating. "The trial concluded when the mouse placed four paws on one of the open arms or stayed in the enclosed arm for a maximum of 300 seconds (avoidance criterion) after leaving the enclosed arm". However, two sentence later "When the mouse extended its body from the enclosed arm and placed one, two, or three paws on one of the open arms before returning to its original position, it was considered as a trial." So, these two definitions of a trial are not in agreement. Later, there is a formula with "tries". However, "tries" are not defined.

Response: Thank you for your observation. We apologize for this confusion. We have clarified this paragraph for a better understanding (page 5, lines 198-207).

(20) Line 219: "Watson53"

Response: Thank you for your observation. We have corrected this typo and changed "Watson53" to Watson [54] (page 5, line 223).

(21) Figure 1: These data are unbelievable. With a group size of 9, there is more or less almost no variability (SE is SEM, correct?), also after 24h. Actually, it would be nice to see a table with the individual data (or a graph with paired exploration times for the two objects). I guess I never saw such robust data in object recognition.

Response: Thank you for your observation. SEM is correct. We have added a table with the individual recognition index data for object recognition and object-in-context recognition tasks as supplementary material (Table 1).

(22) Line 256: " This value was lower in the aged group." Which value? The value in the sentence before was the differences between the two groups at the two time points.

Response: Thank you for your observation. We apologize for the mistake. We meant recognition index values. We rephrased the paragraph to be more specific (page 6, lines 256-258 and 261-262).

(23) Line 261: ANOVA or t-test?

Response: We apologize for the mistake. For the results of figures 1-3, we performed a t-test. For the results in figures 4-8, we performed an ANOVA test. We also corrected the description of the Materials and Methods section (page 6, lines 249-250).

(24) Figure 5: Vehicle-treated mice are much worse than the untreated mice. There is actually a strong effect of the vehicle administrations, which is rescued by TIB. Can the authors give more details on the oral administrations? When exactly were they done? How much later were the test?

Response: Thanks for your comment. Tibolone was administered orally daily at 12:00 noon for 12 weeks with the aid of a special oral gavage for mice, which ensures that the tibolone dose is deposited in the stomach.

The elevated T-maze test consists of a training session where the mouse can explore the closed arm for 300 seconds. If the mouse leaves the closed arm by placing its four paws on one open arm, the trial is terminated, and the animal is returned to its crate for 30 seconds. After this time, the mouse is again subjected to as many trials as necessary to reach the avoidance criterion; in other words, it has to remain in the closed arm for 300 seconds.

Some animals take up to 4 training sessions and others, up to 6 sessions.

(25) Line 303: "Outpatient"?  (= Ambulation???)

Response: Thank you for your observation. We corrected the translation mistake and replaced “Outpatient” with “ambulation” throughout the manuscript.

(26) Figure 7/8: The blots strongly indicate that not only the protein expression levels of each animal was normalized to GADPH but also to the mean of all groups. Is this correct? If yes, this should be described. If my suggestion is not correct, why are the ratios to high? ChAT expression in aged mice is for example much weaker than GADPH expression. Nevertheless, the ratio is higher ca. 1.25 (which would indicate more ChAT expression than DADPH expression!).

Response: Thank you for your observation. We corrected the description in the Materials and Methods section (page 6, lines 244-245).

(27) Lines 335ff: Somehow, the second sentence does not fit to the first.

Response: Thank you for your observation. We have clarified the paragraph in the Discussion section for a better understanding as follows (page 13, lines 343-345):

As men age, testosterone levels decrease, resulting in symptoms such as asthenia, decreased muscle mass, osteoporosis, and reduced sexual activity. This set of symptoms is known as androgen deficiency in the aging male (ADAM) syndrome.

(28) Lines 347ff: I got lost in this sentence: "The results obtained in the present work clearly show a difference between two age stages, where it was observed that the consolidation of memory is mostly affected in old mice, because in the object recognition and object-in- context tests, a decrease was observed in the group of old mice, this indicating a clear difference between the neurological states that two different age groups undergo, in which the neurodegeneration of different regions of the brain conditions a decrease in cognition in old mice in the recognition of objects- without-context". First, I guess that several parts of the sentence can be deleted since unnecessary. Second, dividing in 2 or 3 sentences would facilitate understanding. Third, why do you think it a problem of consolidation? It also could be retention, right?

Response: Thank you for your observation. We have rephrased the paragraph in the Discussion section for a better understanding of the idea as follows (page 6, lines 355-359):

Our results showed differences between young and old mice. Memory retention and consolidation are mostly affected in aged mice, probably because of the neurodegeneration of different brain regions in these animals. Although recognition of prior information remains relatively stable throughout life, the ability to encode and recall contextual information tends to decline with age.

(29) Line 263: Do mice really learn this avoidance response? Where is this shown? The data more look like this avoidance response is innate, as shown by risk indexes very close to 1.

Response: Thank you for your interesting observation. As far as we know, animals do learn the avoidance response. In fact, that is why De-Mello and Carobrez adapted the elevated T-maze test to analyze memory and not just anxiety. This is achieved as the animals undergo as many training sessions as necessary to meet the criterion of remaining in the closed arm for 300 seconds. These training sessions make a difference with respect to the analysis of anxiety. Therefore, the

risk index close to one represents that the animals do not enter the open arms just because they learned the avoidance response.

  1. De-Mello and A. P. Carobrez, ‘Elevated T-maze as an animal model of memory: effects of scopolamine’, Behavioural Pharmacology, 2002, vol. 13, no. 2, pp. 139–148, doi: 10.1097/00008877-200203000-00005

(39) Line 378f: There are many potential reasons for decreased locomotor activity in aged mice. Loss of muscle mass, worse energy homeostatis, poorer sensory abilities, etc. could be also a potential reason.

Response: We totally agree with the reviewer’s observation. We added this information as suggested (page 13, lines 368-370).

(40) Line 379f: This sentence is incomplete. "In addition, long-term treatment with doses of 0.01 mg/kg"

Response: We apologize for this mistake. We completed and rephrased this sentence for a better understanding of the idea (page 13, lines 383-384).

(41) Line 384f: A verb is missing in this sentence. "Several studies have shown that castrated male rats impair spatial working memory [67–71], decreasing performance in the radial arm maze, in the T-maze, and in object recognition [72]."

(42) Line 386: I don't understand how castration can affect the context.

Response to points 41 and 42: Thank you for your observation. We apologize for the missing verb and the grammar mistake. We corrected and rephrased the paragraph to clarify the idea as follows (page 13, lines 386-389):

Several studies have reported that castrated male rats show decreased spatial working memory, and poor performance in the radial arm maze, T-maze, and object recognition tasks due to drastically decreased testosterone levels.

(43) Line 398: "In the present work, we observed that the ChAT concentration increased in old mice with respect to young mice. " Maybe discuss a possible explanation for this increase.

Response: Thank you for your suggestion. We have included a paragraph to further discuss this result (page 14, lines 400-410):

In the present work, we observed increased ChAT concentrations in old mice with respect to young mice. Therefore, we need to clarify the results of previous studies performed in aged C57BL/6J mice regarding ChAT content. Some studies indicate that ChAT content decreases, but others show that it is not altered, and still others that it increases (Frick et al. 2002). Moreover, increased estrogen receptor expression, particularly estrogen receptor beta (ERβ), has been observed in aged animals (Bean et al. 2014). Similarly, estradiol has been reported to increase ChAT expression in rodent hippocampus (Ferrini et al. 2002). These results could explain our findings regarding increased ChAT expression in old mice relative to young mice. Thus, increased ERβ expression could influence the increased ChAT content in aged animals, although further research is required to confirm these observations.

Frick KM, Burlingame LA, Delaney SS, Berger-Sweeney J. Sex differences in neurochemical markers that correlate with behavior in aging mice. Neurobiol Aging. 2002 Jan-Feb;23(1):145-58. doi: 10.1016/s0197-4580(01)00237-8

Bean LA, Ianov L, Foster TC. Estrogen receptors, the hippocampus, and memory. Neuroscientist. 2014 Oct;20(5):534-45. doi: 10.1177/1073858413519865

Ferrini M, Bisagno V, Piroli G, Grillo C, González Deniselle MC, De Nicola AF. Effects of estrogens on choline-acetyltransferase immunoreactivity and GAP-43 mRNA in the forebrain of young and aging male rats. Cell Mol Neurobiol. 2002 Jun;22(3):289-301. doi:10.1023/a:1020767917795

Round 2

Reviewer 2 Report

Comments and Suggestions for Authors

The manuscript has improved a lot.

For me, still the extremely low variability in the behavioral data are hard to believe and I assume that readers will feel similarly. Therefore, also following the idea of open science, I would like to suggest that the videos of the experiments are put online. For the table with the behavioral data, I would like to suggest to add the actual measured values (exploration behavior) and not only to show the index data.

Also, it would be helpful to see how the observations by the two blind reviewers correlate with each other.

Comments on the Quality of English Language

Much better now. It would be good if the editorial office has an eye on language during final editing of the article.

Author Response

 We thank the referees for carefully reviewing the manuscript and their opinions regarding its scientific content and presentation. In what follows, the reviewers’ comments are responded to in blue. 

Reviewer #1 

For me, still the extremely low variability in the behavioral data are hard to believe and I assume that readers will feel similarly. Therefore, also following the idea of open science, I would like to suggest that the videos of the experiments are put online. For the table with the behavioral data, I would like to suggest to add the actual measured values (exploration behavior) and not only to show the index data. Also, it would be helpful to see how the observations by the two blind reviewers correlate with each other. 

Response: 

Thank you for your comment, we have attached table 2 with the data of the exploration time of each animal, of the two blind observers, where you can see that the data of both observers are very similar. 

We have also attached a representative video of each group (Young 1 h, Aged 1 h, Young 24 h, Aged 24 h) of the object recognition test and the object recognition in context test.
